# Sperm physiology and in vitro fertilising ability rely on basal metabolic activity: insights from the pig model

Yentel Mateo-Otero [1,2,5], Francisco Madrid-Gambin [3,5], Marc Llavanera[1,2], Alex Gomez-Gomez[3], Noemí Haro[3], Oscar J. Pozo [3,6✉] & Marc Yeste [1,2,4,6✉]

Whether basal metabolic activity in sperm has any influence on their fertilising capacity has not been explored. Using the pig as a model, the present study investigated the relationship of energetic metabolism with sperm quality and function (assessed through computer-assisted sperm analysis and flow cytometry), and fertility (in vitro fertilisation (IVF) outcomes). In semen samples from 16 boars, levels of metabolites related to glycolysis, ketogenesis and Krebs cycle were determined through a targeted metabolomics approach using liquid chromatography-tandem mass spectrometry. High-quality sperm are associated to greater levels of glycolysis-derived metabolites, and oocyte fertilisation and embryo development are conditioned by the sperm metabolic status. Interestingly, glycolysis appears to be the preferred catabolic pathway of the sperm giving rise to greater percentages of embryos at day 6. In conclusion, this study shows that the basal metabolic activity of sperm influences their function, even beyond fertilisation.

[1] Biotechnology of Animal and Human Reproduction (TechnoSperm), Institute of Food and Agricultural Technology, University of Girona, ES-17003 Girona, Spain. [2] Unit of Cell Biology, Department of Biology, Faculty of Sciences, University of Girona, ES-17003 Girona, Spain. [3] Applied Metabolomics Research Group, Hospital del Mar Medical Research Institute (IMIM), ES-08003 Barcelona, Spain. [4] Catalan Institution for Research and Advanced Studies (ICREA), ES-08010 Barcelona, Spain. [5] These authors contributed equally: Yentel Mateo-Otero, Francisco Madrid-Gambin. [6] These authors jointly supervised this work: Oscar J. Pozo, Marc Yeste. ✉email: opozo@imim.es; marc.yeste@udg.edu

The spermatozoon has historically been regarded as a mere vehicle to deliver the paternal genome into the oocyte; consequently, the importance of paternal factors for oocyte fertilisation, embryo development and, even, offspring health, has been traditionally overlooked. This has led to the misconception that the male contribution to early embryo development solely relies on the sperm genome in terms of DNA integrity and epigenetic signatures[1]. In the last decades, however, multiple studies demonstrated that sperm proteome (reviewed in ref. [2]), lipidome (reviewed in ref. [3]) and transcriptome (reviewed in ref. [4]) also have a crucial influence on oocyte fertilisation and embryo development in mammals. Despite the fact that the sperm metabolome has been proven to affect male fertility in several mammalian species (reviewed in ref. [5]), the repercussion of sperm bioenergetics on oocyte fertilisation and embryo development remains unknown.

Mammalian sperm consume ATP for a wide range of functions, including capacitation, hyperactivation, acrosome reaction and oocyte penetration, each occurring in different environments. Upon ejaculation, sperm come into contact with seminal plasma (SP) and, when deposited within the female reproductive tract, they interact with uterine and oviductal fluids, which are known to differ in terms of ion[6] and metabolite[7–9] composition. Changes in these surrounding biofluids and, therefore, the availability of substrates and oxygen, together with the dramatic changes that sperm undergo to become fertilising competent[10], force these cells to utilise diverse metabolic pathways, including glycolysis and oxidative phosphorylation (Oxphos)[11,12] to satisfy their energetic requirements. In addition, the metabolic pathway preferentially used by sperm is highly species-specific[12]. While glycolysis appears to be the main energetic pathway in humans and rodents[13,14], Oxphos seems to be predominant in horses[15], and a balance of both occurs in cattle[16]. There is, notwithstanding, a discrepancy in the metabolic pathway preferred by pig sperm[13,17]. Remarkably, the diversity of conditions to which sperm are subject to during their life cycle, along with the species-specific differences in their metabolism, contribute to the ongoing debate surrounding the catabolic pathways used by these cells[18].

Because, in sperm, mitochondrial membrane potential, an indirect method to estimate energy production, has been found to affect in vitro fertilisation (IVF) outcomes[19–22], it is reasonable to hypothesise that energetic-related metabolites, e.g., metabolites linked to glycolysis, ketogenesis and Oxphos, are also involved. In this sense, targeted metabolomics approaches offer the possibility of exploring cellular metabolism and metabolic status under specific conditions. Thus, by identifying metabolites in biological samples such as fluids or cell extracts, one can describe physiological processes, generate new hypotheses for unsolved metabolic interrogations and even find potential biomarkers[23]. Against this background, the present study aimed to address: (1) the metabolic pathway preferred by porcine sperm to produce energy; 92) whether the energetic metabolic state of sperm is related to their quality and function; and (3) the potential relationship between sperm energetic metabolism and oocyte fertilisation and subsequent embryo development. In addition, the pig has been proposed as a suitable animal model, not only on the grounds of the physiological similarities with humans and the availability of semen samples of high volume but also because rodent species may not be appropriate for sperm physiology studies, as epididymal sperm are never in contact with SP[24]. Yet, before their use for metabolomic studies, the energetic pathway utilised by pig sperm and how similar this is to their human counterparts needs to be elucidated. To this end, sperm samples were split into three aliquots. Two were used to evaluate sperm quality/function parameters and run IVF experiments, respectively. The other was intended to quantify sperm intracellular metabolites related to glycolysis, ketogenesis, polycarboxylic acids cycle and Oxphos through liquid chromatography-tandem mass spectrometry (LC-MS/MS), which provides highly specific, sensitive, accurate and reproducible results[25].

## Results

**Dimensionality reduction**. Three different blocks were included to assess sperm physiology and in vitro fertility: sperm quality, sperm function and IVF outcomes. Parameters included in the sperm quality block were the percentage of sperm with normal morphology, the percentage of motile sperm (total motility), the percentage of sperm with progressive motility (progressive motility) and the percentage of viable sperm. The first principal component (PC) from principal component analysis (PCA) of this block represented up to 77% of total variability. The sperm function block was comprised of three different variables: the percentage of viable sperm with an intact acrosome, intracellular calcium levels and the percentage of sperm with high mitochondrial membrane potential; one outlier was detected and excluded from this block. In this case, the first PC explained 46% of the total variability. The IVF outcomes block encompassed fertilisation rate at day 2 post-fertilisation and the different embryo developmental stages evaluated 6 days after fertilisation, which included: percentages of morulae, early blastocysts/blastocysts and hatched/hatching blastocysts; the sum of morulae, early blastocysts/blastocysts and hatched/hatching blastocysts; and the total number of embryos. Two additional ratios were also calculated: the developmental potential of embryos at day 6 and the developmental competency of fertilised embryos. The PCA of this dataset showed 50% of the total variability in the first PC. Additional information is provided in Supplementary Table 1.

**Association of metabolic signature with sperm physiology and IVF outcomes**. The Partial Least Squares (PLS) model denoted an association between metabolites and sperm physiology and in vitro fertility reduced dimension feature vectors. The performance of the different models showed a cross-validated $R^2$ of 0.823, 0.830 and 0.460 for sperm quality, sperm function and IVF outcomes, respectively. Likewise, predictive $Q_2$ values of 0.269, 0.693 and 0.387 were obtained for the aforementioned blocks (Fig. 1). The predictive capability of these models was validated with a permutation test ($P$ value of 0.013, <0.001 and <0.001 for sperm quality, sperm function and IVF outcomes, respectively; Fig. 1). The included feature selection of the model exhibited several metabolites associated to each block, as shown in Table 1. Overall, citrate, isocitrate, lactate, citrate/lactate and citrate/malate were found to be linked to sperm quality, sperm function and IVF outcomes. In addition, α-hydroxyglutarate/citrate, α-hydroxyglutarate/isocitrate and α-ketoglutarate/isocitrate were also related to both sperm quality and sperm function. Finally, acetoacetate, fumarate and isocitrate/citrate were found to be correlated with sperm quality.

**Multi-block data analysis reveals specific metabolic processes associated with sperm physiology and IVF variables**. Once metabolic markers associated with sperm physiology and IVF inputs were detected, whether these observations were intercorrelated across the dataset was investigated. For this purpose, the relationships between features observed from the N-integration with Projection to Latent Structures model across data were examined and visualised in an integrative network, as shown in Fig. 2. For subsequent analyses, the sample containing the previously detected outlier in the PCA of the sperm function block was excluded. The correlation of latent components between blocks is summarised in Supplementary Table 2.

Multi-block data analysis revealed that not only were sperm physiology and in vitro fertility blocks closely related to the metabolomics one but also with each other. Unfolded pair-wise

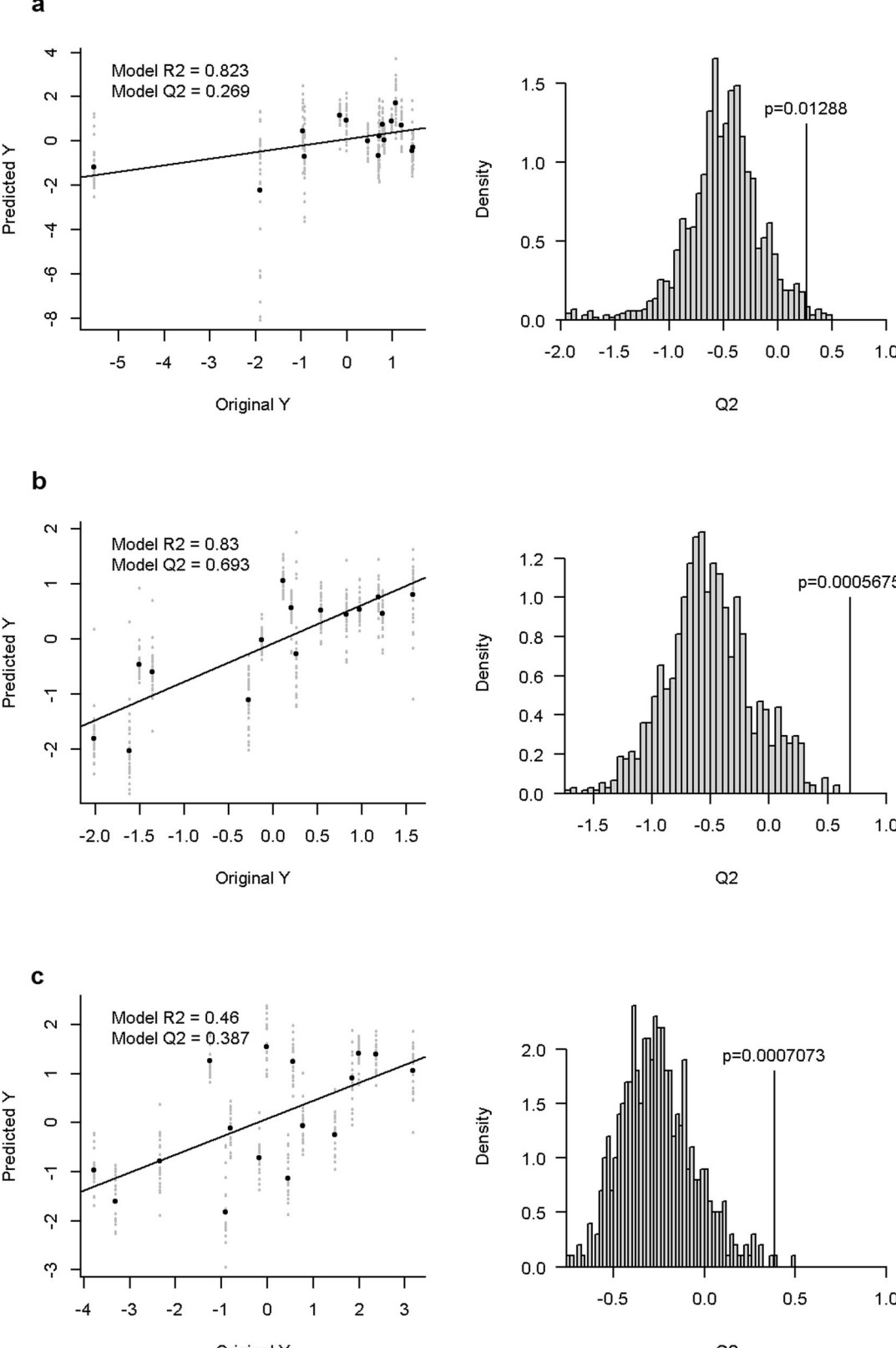

**Fig. 1 Partial least square (PLS) regression plots of actual and predicted. a** sperm quality ($n = 16$), **b** sperm function ($n = 15$) and (**c**) in vitro fertilisation outcomes ($n = 16$). Reduced-dimension feature vectors from projection are shown in the left, and permutation tests based on the prediction capability are depicted on the right. In the x-axis, PLS regression plots show original centred, reduced-dimension feature vectors from each principal component analysis. The values predicted by the models are displayed on the y axis. The slope is defined by the prediction capability values (Q2) tested through permutation tests. The vertical lines of permutation test plots indicate the Q2 values obtained in each block.

**Table 1 Metabolites associated with sperm physiology and in vitro fertility in the PLS models.**

| | Metabolites | Recursive rank[a] | LR[b] | P value[c] | FDR | Beta[c] |
|---|---|---|---|---|---|---|
| Sperm quality | Acetoacetate | 3.64 | 1 | <0.001 | 0.002 | 0.04 |
| | Lactate | 5.52 | 2 | 0.006 | 0.035 | <0.01 |
| | Citrate | 7.73 | 3 | 0.991 | 0.991 | <0.01 |
| | Citrate/malate | 8.26 | 4 | 0.343 | 0.573 | <0.01 |
| | Fumarate | 8.37 | 5 | 0.008 | 0.036 | 0.01 |
| | α-hydroxyglutarate/isocitrate | 9.81 | 6 | 0.781 | 0.879 | <0.01 |
| | Isocitrate | 9.81 | 7 | 0.056 | 0.126 | 0.01 |
| | Citrate/lactate | 9.96 | 8 | 0.350 | 0.573 | <0.01 |
| | α-Ketoglutarate/Isocitrate | 12.76 | 9 | 0.529 | 0.793 | <0.01 |
| | Isocitrate/citrate | – | – | <0.001 | 0.002 | 9.77 |
| Sperm function | Citrate | 2.03 | 1 | <0.001 | 0.002 | −1.27 |
| | Citrate/lactate | 2.05 | 2 | <0.001 | 0.001 | −1.35 |
| | Citrate/malate | 3.82 | 3 | <0.001 | 0.005 | −1.27 |
| | α-hydroxyglutarate/isocitrate | 6.53 | 4 | 0.005 | 0.018 | 1.25 |
| | α-Ketoglutarate/isocitrate | – | – | 0.005 | 0.018 | 1.40 |
| | Isocitrate | – | – | 0.019 | 0.058 | −1.39 |
| IVF outcomes | Citrate/lactate | 3.37 | 1 | <0.001 | 0.003 | −0.32 |
| | Citrate/malate | 3.41 | 2 | 0.003 | 0.029 | −0.42 |
| | Citrate | 4.67 | 3 | 0.007 | 0.041 | −0.37 |
| | α-Ketoglutarate | 9.71 | 4 | 0.092 | 0.251 | −0.42 |
| | Isocitrate | 10.25 | 5 | 0.050 | 0.181 | −0.26 |
| | Lactate | – | – | 0.044 | 0.181 | 0.29 |

*FDR* false discovery rate.
[a]Recursive rank of double-cross-validation PLS regression. In each block, metabolites were repeatedly ranked in each outer iteration and cumulated in the recursive rank feature. Only significant variables are numbered.
[b]Loading rank displays the absolute ranking of variables based on their importance.
[c]P and beta values of linear models. P values were corrected utilising the Benjamini–Hochberg formula.

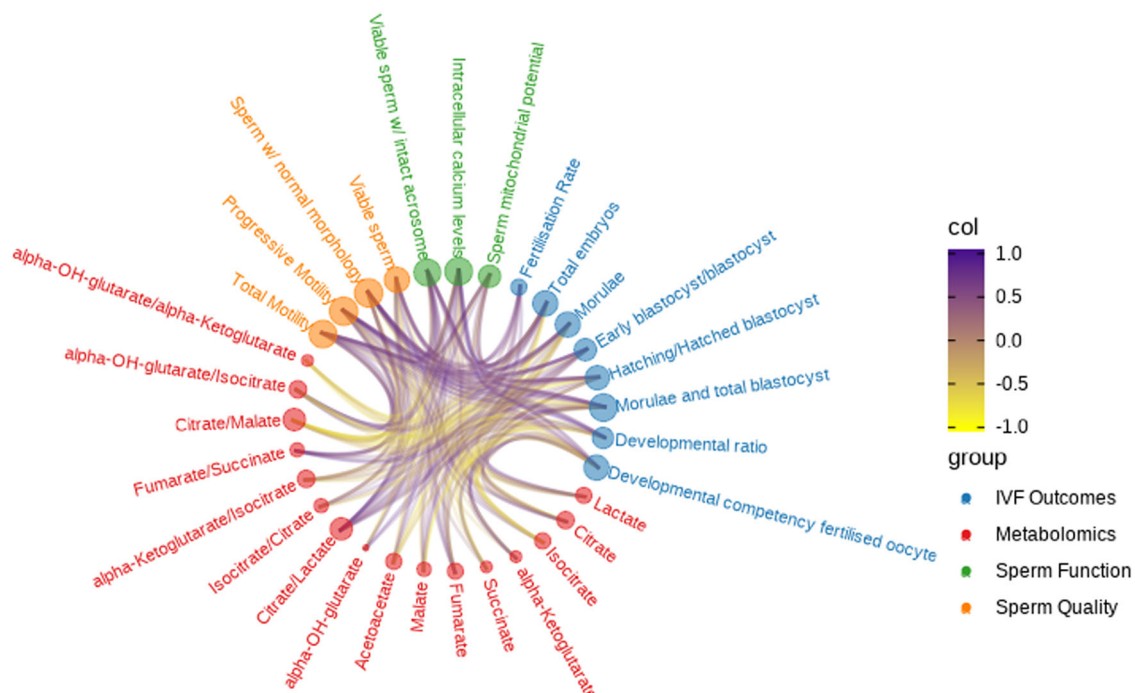

**Fig. 2 Integrative network graph depicting correlations derived from the N-integration with projection to latent structures between blocks.** In vitro fertilisation outcomes (blue), metabolomics (red), sperm function (green) and sperm quality (orange) were associated on the basis of a similarity score > | 0.3| (*n* = 15). Lines are coloured according to similarity scores: positive associations are shown in purple, whereas inverse associations are depicted in yellow. Nodes (circles) represent variables and are sized according to the number of connections. Further information can be found in Supplementary Table 3. col colour, IVF in vitro fertilisation, OH hydroxy.

similarity scores from multi-block data integration are shown in Supplementary Table 3. The general trend was that whereas citrate, isocitrate and citrate/malate showed a positive association with sperm quality and in vitro fertility blocks, α-hydroxyglutarate/isocitrate, α-hydroxyglutarate/α-ketoglutarate and α-ketoglutarate/citrate exhibited a negative relationship. Interestingly, these associations were found to be inverse in the case of sperm function parameters.

Focusing on sperm quality, the most relevant positive relationships were noted between (i) the percentage of sperm with progressive motility and citrate, citrate/lactate and citrate/malate (similarity score >0.70); (ii) the percentage of sperm with normal morphology and citrate, citrate/lactate and citrate/malate (similarity score >0.70); and (iii) the percentage of viable sperm and citrate, citrate/lactate and citrate/malate (similarity score >0.50). On the other hand, scarce negative relationships were observed between sperm quality parameters and metabolites. The most relevant interactions were found between (i) the percentage of sperm with progressive motility and α-ketoglutarate/isocitrate and α-hydroxyglutarate/isocitrate (similarity score = −0.67 and −0.58, respectively); and (ii) the percentage of sperm with normal morphology and α-hydroxyglutarate/isocitrate (similarity score = −0.61).

Regarding sperm function, a strong positive relationship (similarity score >0.70) was seen between (i) the percentage of sperm with high mitochondrial membrane potential and α-hydroxyglutarate/α-ketoglutarate; (ii) intracellular calcium levels and α-hydroxyglutarate/isocitrate and α-ketoglutarate/isocitrate; and (iii) the percentage of viable sperm with an intact acrosome and α-hydroxyglutarate/α-ketoglutarate. Moreover, strong negative relationships were also detected between (i) intracellular calcium levels and citrate, citrate/malate and citrate/lactate (similarity score < −0.80); (ii) the percentage of sperm with high mitochondrial membrane potential and isocitrate/citrate, fumarate/succinate and α-ketoglutarate (similarity score < −0.70); and (iii) the percentage of viable sperm with an intact acrosome and α-ketoglutarate, isocitrate/citrate and isocitrate (similarity score < −0.70).

Finally, several strong relationships between different metabolites and IVF outcomes were found. Fertilisation rate at day 2 was the variable with the fewest associations with the other parameters, showing a negative relationship with isocitrate/citrate and fumarate/succinate (similarity score < −0.75), and a positive relationship with α-hydroxyglutarate, succinate/α-ketoglutarate and α-hydroxyglutarate/α-ketoglutarate (similarity score >0.50). Regarding embryo development parameters evaluated at day 6, several relevant relationships were also identified. Specifically, positive correlations were observed between (i) the percentage of total embryos and citrate, citrate/lactate and citrate/malate (similarity score >0.70); (ii) the percentage of morulae plus blastocysts and citrate/lactate, citrate and citrate/malate (similarity score >0.75); (iii) the percentage of morulae and citrate and isocitrate (similarity score >0.70); (iv) the percentage of hatching and hatched blastocysts and citrate, citrate/malate and citrate/lactate (similarity score >0.65); and (v) the developmental competency of fertilised oocytes and citrate and isocitrate (similarity score >0.70). On the other hand, negative correlations between (i) the percentage of morulae and α-hydroxyglutarate/isocitrate (similarity score < −0.75); (ii) the percentage of early blastocysts plus blastocysts and isocitrate/citrate and fumarate/succinate (similarity score < −0.75); (iii) the percentage of morulae plus blastocysts and α-ketoglutarate/isocitrate (similarity score < −0.75); and (iv) the developmental competency of fertilised oocytes and α-hydroxyglutarate/isocitrate (similarity score < −0.75) were also noticed.

## Discussion

The precise catabolic pathway preferred by the sperm of each species is, in some cases, controversial. There are discrepancies between studies as, among other factors, the composition of semen extenders/preservation media differs. In effect, the availability of substrates directly influences the energetic pathway preferentially used by sperm cells. Particularly in pigs, while a recent study reported that non-capacitated sperm heavily rely upon mitochondrial Oxphos for ATP production[17], their

metabolism has traditionally been assumed to be preferentially glycolytic. This assumption resides on the fact that, in porcine, (i) sperm catabolism of glucose produces lactate[26]; (ii) sperm contain mitochondria with few and practically nonvisible inner membrane crests[27]; (iii) sperm mitochondria crests are less condensed compared to their horse counterparts (whose metabolism is mainly oxidative)[28]; and (iv) a specific lactate dehydrogenase isozyme is present in sperm[29]. The results presented here, specifically the identification of lactate in sperm lysates and the absence of strong positive associations between sperm physiology and Krebs cycle intermediate metabolites, corroborate that glycolysis can be regarded as the main catabolic pathway used by non-capacitated pig sperm in high-quality, fertile semen samples. Interestingly, the data collected in this work confirm that the strategy of pig sperm to produce energy resembles to that of their human counterparts[13,14]. These metabolism similarities, together with the already reported analogy between species in terms of sperm physiology[24], opens a new range of possibilities in the study of the influence of metabolism on sperm function, fertility potential and contribution to embryo development using the pig as an animal model. For instance, fertility potential of sperm has been widely reported to be affected by their molecular composition, which mainly includes the proteome[2], lipidome[3], transcriptome[4] and metabolome[5]. Although the sperm metabolic profile is known to affect in vivo fertility outcomes in mammals[5], the exact way through which this element affects fertility is yet to be uncovered. The next step in this study was, therefore, to address how the energetic metabolic signature in mammalian sperm shapes their function and affects IVF outcomes.

Not only do sperm quality and function involve the evaluation of conventional sperm quality parameters such as motility, viability and morphology, but also that of other physiological processes, such as the acrosome reaction, mitochondrial activity and calcium homeostasis, among others[30]. The aforementioned processes are related to the sperm ability to fertilise an oocyte, therefore they have been traditionally used to estimate the reproductive performance of semen samples[30]. Characterising the factors that might be influencing sperm quality and function is thus crucial to understand the molecular mechanisms underlying the fertility of the male gamete. For this reason, the present work sought to address whether the energetic metabolic signature of sperm is related to their quality, function and fertilising ability. The positive relationship observed in the PLS model between lactate and citrate—but not other Krebs cycle metabolites—and sperm quality indicates that the main catabolic pathway in non-capacitated sperm samples of good quality (high motility and viability, and low incidence of morphological abnormalities) is glycolysis. This was further confirmed by multi-block data analysis, which revealed: (i) a positive relationship between glycolysis intermediates and the percentage of sperm with progressive motility, the percentage of viable sperm and the percentage of sperm with normal morphology; and (ii) a negative relationship between Krebs cycle intermediates and the percentage of sperm with progressive motility and the percentage of sperm with normal morphology. Although mitochondrial respiration is the most efficient source of ATP, glycolysis has also been associated with sperm of good quality in other mammalian species. In effect, glycolysis has been reported to be strongly related to sperm viability and motility in cattle[31], mice[13,32,33] and humans[34,35]. Yet, while glycolysis seems to be crucial for sustained sperm quality, additional studies should be carried out to evaluate if the mitochondrial activity also contributes to maintain sperm quality and even plays an essential role in the regulation of the events occurring in the female tract.

The PLS model showed that sperm function was negatively related to citrate, citrate/lactate, citrate/malate and isocitrate

(glycolysis metabolic markers), and positively related to α-hydroxyglutarate/isocitrate and α-ketoglutarate/isocitrate (Oxphos metabolic markers). From the multi-block analysis, the most interesting relationship identified was between Oxphos intermediates and intracellular calcium levels in sperm. The elevation of intracellular calcium levels is one of the first events of capacitation, a process physiologically induced in the female reproductive tract by the means of which sperm become fertilising competent[36]. In this sense, considering that capacitation should not occur in non-capacitating media, high levels of intracellular calcium in sperm could be understood as an indicator of poorer quality samples and, probably, reproductive outcomes as by the time of fertilisation, the status of the cell would not be appropriate. Thus, the positive association of α-ketoglutarate/citrate and α-hydroxyglutarate/isocitrate with intracellular calcium levels would line up with the previously set hypothesis: glycolysis rather than Oxphos is related with the best sperm quality traits. In addition, the association between intracellular calcium levels and Oxphos could also be considered to open a new research question: does porcine sperm metabolism switch during capacitation as already observed in rodents[37]? Ramió-Lluch et al.[38] already suggested that capacitation and the acrosome reaction are accompanied by a progressive increase of mitochondrial activity. In spite of this, as intracellular calcium levels are only the first step of capacitation, further research should determine the changes in the energetic metabolic signature occurring during sperm capacitation and how female fluids can affect sperm metabolism during these events. On the other hand, interestingly, the present work also found that the percentage of sperm with an intact acrosome was negatively related to glycolysis. Not much research about the role of sperm metabolism on physiological/spontaneous acrosome reaction has been conducted. A recent publication, however, reported that spontaneous acrosome reaction is independent from metabolic pathways in bovine sperm[39]. Whether spontaneous acrosome reaction in porcine sperm is also independent from metabolism is unknown. Since the present study was a first attempt to determine how metabolism affects sperm physiology, further studies are needed before a firm conclusion can be drawn.

The utility of metabolomic technologies to predict in vivo fertility from SP[40–47] or sperm[5,48] has widely been proved in several mammalian species. To the best of the authors' knowledge, nevertheless, no study in any mammalian species has looked into the relationship between the sperm metabolome and IVF outcomes. In general, the positive relationship observed in the PLS model between the first Krebs cycle metabolites (citrate, isocitrate and α-ketoglutarate) and IVF outcomes indicates that samples with the best IVF outcomes are highly associated to sperm whose main catabolic pathway is glycolysis. This would be in agreement with previous findings in mice, where the knockout of genes encoding for glycolytic-related proteins, such as Glyceraldehyde 3-phosphate dehydrogenase-S (*Gapdh*)[13], Phosphoglycerate kinase 2 (*Pgk2*)[33], Lactate dehydrogenase C (*Ldhc*)[32] or Cytochrome C (*CytCt*)[49], revealed that glycolysis rather than Oxphos is essential to preserve male fertility. Considering these results, whether sperm metabolism modulates the sperm ability to fertilise the oocyte and/or contributes to the subsequent embryo development was interrogated. The multi-block data integration showed a moderate, positive relationship between fertilisation rate and Oxphos, and a strong positive association between the total number of embryos at day 6 and glycolysis. This, together with the fact that the current study found a very strong association between fertilisation rate and the percentage of sperm with high mitochondrial membrane potential, would indicate that, at first glance, sperm using Oxphos as the principal catabolic pathway would have greater oocyte fertilising ability. Yet, oocytes fertilised by sperm preferentially using glycolysis appeared to produce more embryos at day 6. In a similar

fashion, the developmental competency of fertilised oocytes was found to be positively associated to glycolysis-related metabolites. These results thus suggest that pre-implantation embryo development, rather than oocyte fertilisation, is closely influenced by sperm glycolysis. This idea would be supported by an additional finding of this study: sperm with the lowest levels of Oxphos metabolites are those that led to the highest percentages of the most developed embryos (i.e., percentages of morulae and blastocysts). How sperm metabolism can condition embryo development is unknown, but reactive oxygen species (ROS) might hold the key. Excessive ROS are known to affect sperm physiology through lipid peroxidation, motility reduction, apoptosis-like changes and even DNA damage[50]. Focusing on the latter, it has recently been reported that sperm DNA damage negatively affects embryo development in pigs[51]. Considering that ROS are mainly produced as a result of cellular respiration, one explanation for the data collected in the present study would be that sperm with excessive Oxphos activity could also bear greater DNA damage, which would compromise their IVF outcomes. Thus, sperm using glycolysis as their main energy source would probably contain less DNA damage, which would allow embryos to reach further pre-implantation stages. As this is only a hypothesis, the relationship between Oxphos/glycolysis and sperm DNA damage should be studied in the future.

The targeted metabolomics approach taken in this work allowed the characterisation of the main catabolic pathway in non-capacitated pig sperm, and addressed the relationship between their energetic metabolic status and fertility outcomes. This study also supported that glycolysis rather than Oxphos is used by sperm samples with good quality to produce energy. In addition, embryo development seemed to be tightly associated to glycolysis-related metabolites. These findings are a first steppingstone to explain how the sperm metabolome may influence fertility, as it shows that sperm metabolism has an impact on IVF outcomes. In addition, taking into consideration the similarities between pigs and humans in terms of the catabolic pathway preferred by their sperm, the results shown herein could be useful for the estimation of the success of IVF cycles using a non-invasive approach. Forthcoming studies should thus be focused on setting up specific metabolic biomarkers that could predict reproductive success.

## Methods

**Reagents.** All reagents used in the present study were purchased from Sigma (Merck, Darmstadt, Germany) unless stated otherwise.

**Animals.** Semen samples were provided by an artificial insemination (AI) centre (Gepork S.L.; Masies de Roda, Spain), which follows the ISO certification (ISO-9001:2008), the EU Directive 2010/63/EU for animal experiments, the Catalan Animal Welfare Law, and the current regulation on Health and Biosafety issued by the Department of Agriculture, Livestock, Food and Fisheries, Regional Government of Catalonia, Spain. As ejaculates were commercially acquired from an AI centre and animals were not manipulated for the sole purpose of the present experiment, permission from an Ethics Committee was not required.

Ejaculates from healthy and sexually mature Pietrain boars (1–3 years old) were collected between June and July 2021 using the gloved-hand method. Immediately after collection, semen samples were diluted to a final concentration of $33 \times 10^6$ sperm/mL using a commercial extender (Vitasem LD, Magapor S.L., Zaragoza, Spain), and stored at 17 °C for 24 h. Upon arrival at the laboratory, the semen quality of doses was assessed to check if they met the conventional minimum requirements (sperm viability greater than 80% and motility greater than 70%).

On the other hand, ovaries were recovered from pre-pubertal gilts sacrificed for food purposes at a local abattoir (Frigorífics Costa Brava; Riudellots de la Selva, Girona, Spain).

**Experimental design.** Sixteen ejaculates meeting the quality standards and coming from the AI centre (each came from a separate boar; i.e., 16 boars) were included in the present study, and split into three aliquots. The first was used to assess sperm quality (which included sperm motility, viability and morphology) and function (which included acrosome integrity, intracellular calcium levels and mitochondrial membrane potential), and the second was used for IVF experiments. In brief, a

total of 650 oocytes were matured, fertilised, and both the fertilisation rate (day 2) and rates of embryo development at different pre-implantation stages (day 6) were recorded. Finally, the third aliquot was stored at −80 °C and later served to investigate sperm metabolomics through LC-MS/MS.

### Sperm quality evaluation

*Sperm motility*. Semen samples were pre-warmed at 38 °C for 15 min, and 3 µL was placed into a Leja20 counting chamber (Leja Products BV; Nieuw-Vennep, The Netherlands). Following this, samples were evaluated under an Olympus BX41 microscope (Olympus; Tokyo, Japan) with a negative phase-contrast objective (Olympus 10× 0.30 PLAN objective, Olympus), through a computer-assisted sperm analysis (CASA) system (Integrated Sperm Analysis System, ISAS V1.0; Proiser S.L.; Valencia, Spain). Two technical replicates were evaluated per sample, and at least 1000 sperm were examined in each replicate.

Two different parameters were recorded: the percentage of motile sperm, which considered those motile sperm whose average path velocity (VAP) was ≥10 µm/s; and the percentage of sperm with progressive motility, which included those motile sperm that exhibited a percentage of straightness (STR) ≥ 45%.

*Sperm morphology*. Sperm morphology was evaluated after dilution in 0.12% formaldehyde saline solution (PanReac AppliChem; Darmstadt, Germany; 1:1, v:v). Samples were observed under a phase-contrast microscope at ×1000 magnification (Nikon Labophot; Nikon; Tokyo, Japan), and 200 sperm cells were examined. Sperm cells were graded as morphologically normal, or with primary or secondary alterations[30]. The percentage of normal sperm was calculated from those without morphological alterations.

*Sperm viability assessment*. Sperm viability was assessed following the protocol of ref. [52], which uses SYBR-14 that stains sperm nuclei, and propidium iodide (PI) that only labels sperm having a compromised plasma membrane integrity. Briefly, semen samples were adjusted to a final concentration of $4 \times 10^6$ sperm/mL in 1× phosphate-buffered saline (PBS). Next, samples were incubated for 15 min at 38 °C with SYBR-14 (final concentration: 32 nM) and PI (final concentration: 7.5 µM). Stained cells were analysed using a CytoFLEX cytometer (Beckman Coulter; Brea, CA, USA), where SYBR-14 fluorescence was detected by the fluorescein isothiocyanate (FITC) channel (525/40), and PI using the PC5.5 channel (690/50). Both fluorochromes were excited with a 488-nm laser and no spill compensation was applied. Two technical replicates of at least 10,000 sperm were analysed at a constant flow rate, laser voltage and sperm concentration. The percentage of viable sperm corresponded to the SYBR-14⁺/PI⁻ population, after subtracting the percentage of debris particles in the analysis (Supplementary Fig 1).

### Sperm function assessment

Sperm function was determined through the evaluation of intracellular calcium levels, acrosome membrane integrity and mitochondrial membrane potential using a CytoFLEX cytometer. Forward (FS) and side scatter (SS) were measured and linearly recorded for all particles. Subcellular debris and cell aggregates were excluded, and sperm events were positively gated through the adjustment of the analyser threshold on the FS channel. Finally, sperm-specific events were validated on the basis of FS/SS distributions (Supplementary Fig. 1).

Sperm intracellular calcium levels were assessed following ref. [53]; sperm were stained with Fluo3-AM (final concentration: 1.2 µM) and PI (final concentration: 5.6 µM) for 10 min at 38 °C in the dark. Fluo3 was detected through the FITC channel (525/40). The mean of Fluo3 fluorescence intensity per sperm (Fluo3⁺/PI⁻) was recorded and used for subsequent statistical analyses.

Acrosome membrane integrity was evaluated following the protocol of ref. [54], in which sperm were stained with PNA-FITC (final concentration: 1.2 µM) for 5 min at 38 °C in the dark, and then with PI (final concentration: 5.6 µM) for 5 min at 38 °C in the dark. PNA-FITC was detected by the FITC channel (525/40). The percentage of viable sperm with an intact acrosome membrane (PNA-FITC⁻/PI⁻) was recorded and used for subsequent statistical analyses.

Mitochondrial membrane potential was evaluated following the protocol set by Ortega-Ferrusola et al.[55]. Sperm were incubated with JC-1 (final concentration: 750 nmol/L) for 30 min at 38 °C in the dark. In cells with high mitochondrial membrane potential, JC-1 aggregates and emits orange fluorescence, which is collected through the PE channel. On the contrary, in cells with low mitochondrial membrane potential, JC-1 is found in its monomeric form and generates green fluorescence, which is collected through the FITC channel. The percentage of sperm with high mitochondrial membrane potential was recorded and used for subsequent statistical analyses.

### Oocyte maturation, in vitro fertilisation and embryo culture

Ovaries were transported to the laboratory in 0.9% NaCl supplemented with 70 µg/mL kanamycin at 38 °C. Cumulus oocyte complexes (COC) were retrieved from follicles and selected in Dulbecco's PBS (Gibco, ThermoFisher) supplemented with 4 mg/mL BSA. Only COCs exhibiting a complete and compact cumulus mass were included in the study.

For in vitro maturation of oocytes (IVM), TCM-199 (Gibco) supplemented with 0.57 mM cysteine, 0.1% (w:v) polyvinyl alcohol, 10 ng/mL human epidermal growth factor, 75 µg/mL penicillin-G potassium, and 50 µg/mL

streptomycin sulphate was used. COCs were matured in groups of 40–50 in four-well multi-dishes (Nunc, ThermoFisher; Waltham, MS, USA) containing 500 µL of pre-equilibrated maturation medium supplemented with 10 IU/mL equine chorionic gonadotropin (eCG; Folligon; Intervet International B.V.; Boxmeer, The Netherlands) and 10 IU/mL human chorionic gonadotropin (hCG; Veterin Corion; Divasa Farmavic S.A.; Gurb, Barcelona, Spain). After 20–22 h, oocytes were transferred into 500 µL fresh, pre-equilibrated IVM medium without hormones.

Next, mature oocytes were placed in 50-µL drops of pre-equilibrated IVF medium (Tris-buffered medium[56]) containing 1 mM caffeine. Semen samples were adjusted to 1000 sperm per oocyte in IVF medium and, thereafter, oocytes and sperm were co-incubated for 5 h in the incubator; a total of 40 oocytes per ejaculate were inseminated. Potentially fertilised oocytes were subsequently washed and transferred into 500 µL NCSU23 medium[57] supplemented with 0.4% BSA, 0.3 mM pyruvate and 4.5 mM lactate for embryo in vitro culture (IVC). After 2 days, cleaved embryos were counted to calculate fertilisation rates, and then transferred into NCSU23 medium supplemented with 0.4% BSA and 5.5 mM glucose. At day 6 post-fertilisation, the resulting embryos were classified following ref. [58] criteria. Specifically, the percentages of morulae, early blastocysts/blastocysts, hatching/hatched blastocysts and total embryos (sum of morulae, early blastocysts/blastocysts and hatching/hatched blastocysts) were evaluated. In addition, the sum of morulae, early blastocysts/blastocysts and hatching/hatched blastocysts was also determined to calculate the percentage of embryos in advanced stages. Finally, two different ratios were calculated: (i) the developmental potential at day 6, which corresponded to the percentage of morulae, early blastocysts/blastocysts plus hatched/hatching blastocysts divided by the percentage of 2–8 cell embryos; and (ii) the developmental competency of fertilised embryos, calculated as the ratio between the number of embryos at day 2 and those at day 6.

All procedures (oocyte IVM, IVF, and IVC) were carried out at 38.5 °C under a humidified atmosphere of 5% $CO_2$ in air.

### Metabolomics

*Sperm lysis*. A total of 100 million sperm were lysed in 500 µL of lysis buffer (0.1% SDS 0.1% Triton in PBS). After samples were vortexed for 45 min at 4 °C, lysates were centrifuged at 18,000×*g* and 4 °C for 20 min. Supernatants were recovered and stored at −80 °C until LC-MS/MS analysis was carried out. Two technical replicates per semen sample were processed. In addition, and in order to prepare the blank, all the protocols were applied in parallel to four replicates that did not contain sperm samples.

*LC-MS/MS analysis*. Cell lysates were analysed by adapting a previously reported method for the quantification of polycarboxylic acids[59]. The method involved a derivatisation with o-benzylhydroxylamine, a liquid–liquid extraction with ethyl acetate and LC-MS/MS detection using a selected reaction monitoring mode. A LC-MS/MS system consisting of an Acquity UPLC instrument (Waters Associates, Milford, MA, USA) coupled to a triple quadrupole (TQS Micro, Waters) mass spectrometer was used for the analysis. Lactic acid, citric acid, isocitric acid, α-ketoglutarate, succinic acid, fumaric acid, malic acid, acetoacetate and α-hydroxyglutarate were determined. In addition to the concentration of each metabolite, nine ratios between metabolites with potential information about enzyme activity were calculated. MassLynx software V4.1 (Waters Associates) was used for peak integration and data management.

### Statistics and reproducibility

*Data analysis*. Data preprocessing and statistical analyses were conducted using the R software (version 4.2.0). The sample size for linear regression was calculated using the "pwr.f2.test" function from the "pwr" R package[60]. Missing values were replaced by half of the minimum value within the dataset. The Shapiro-Wilk test was used to assess normality. The metabolomics dataset was log-transformed before modelling.

Sperm physiology and in vitro fertility parameters were classified into three main blocks: sperm quality, sperm function and IVF outcomes, and were analysed separately. Each group was log-transformed and scaled prior to running PCA for dimensionality reduction purposes. PCA disposes an orthogonal projection onto a lower dimensional subspace, which captures the majority of the variance of the dataset[61]. Then, variables of each block were projected onto a few principal loading vectors independently, condensing most of the variability of the original data[62]. Score values from the first PC of each block were utilised as a reduced-dimension feature vector in the response block (Y-block), predicted in function of the metabolomics set (X-block) using a multivariate PLS regression. The generation of the PLS model was carried out through the root mean square error of prediction as metric in a repeated double-cross-validation framework[63], including a recursive ranking based on variable importance in projection and sequential backward feature removal[64]. The whole operation was repeated 20 times for improved coverage of inner and outer segments and modelling performance. The model performance was assessed by means of a permutation test of 500 iterations between permuted models, with a random assignation of the observations, and the actual model obtained. Furthermore, linear models were run on metabolic data using the reduced-dimension feature vectors as response. The Benjamini–Hochberg procedure was carried out on all analyses to control the false discovery rate

(FDR)[65]. Only FDR-corrected *P* values lower than 0.05 were considered as statistically significant.

*Multi-block data integration*. Integration of multiple datasets measured on the same observations was conducted utilising the N-integration with Projection to Latent Structures model[66]. This model was built to assess multi-block correlations between sperm quality, sperm function, IVF outcomes, and metabolomic blocks from the same observational units, using the mixOmics R package v 6.18.1[67]. A pair-wise similarity matrix was constructed from the two correlated latent components obtained through the projection to latent structures method. A relevance network graph was created to describe connections between the four datasets, based on the rule of similarity score ≥0.3[68].

**Reporting summary**. Further information on research design is available in the Nature Portfolio Reporting Summary linked to this article.

## Data availability
The datasets used and/or analysed during this study are available as Supplementary Data.

## Code availability
R studio V4.2.0 was used for all analyses. The sample size for linear regression was calculated using the "pwr.f2.test" function from the "pwr" R package. Repeated double-cross-validation PLS regressions were run using the "MUVR" R package (available at https://gitlab.com/CarlBrunius/MUVR). The "mixOmics" R package was used for PCA and the integration of multi-block data was conducted through "block.pls" function. Codes for sample size calculation, PLS regressions, linear models, and multi-block data integration are available on GitHub (https://github.com/Francisco-madrid-gambin/CodeSharingTechnoSperm).

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

## Acknowledgements

The authors would like to thank Frigorífics Costa Brava (Riudellots de la Selva, Girona) for supplying ovaries. This study was funded by the Ministry of Science and Innovation, Spain (AGL2017-88329-R, FJC2018-035791-I, FPU18/00666 and PID2020-113320RB-I00), the Regional Government of Catalonia, Spain (2017-SGR-1229 and 2020-FI-B-00412), and the Catalan Institution for Research and Advanced Studies (ICREA).

## Author contributions

Conceptualisation: Y.M.-O., F.M.-G., M.L., O.P. and M.Y.; methodology: Y.M.-O., F.M.-G., M.L., A.G.-G. and N.H.; formal analysis and investigation: Y.M.-O., F.M.-G. and M.L.; writing—original draft preparation: Y.M.-O. and F.M.-G.; writing— review and editing: M.L., A.G.-G., O.P. and M.Y.; funding acquisition: O.P. and M.Y.; supervision: O.P. and M.Y. All authors have read and agreed to the published version of the manuscript.

## Competing interests

The authors declare no competing interests.
