## [Peer Review File · Communications Biology]

Reviewers' comments:

Reviewer #1 (Remarks to the Author):

In the manuscript "Sperm physiology and in vitro fertility outcomes rely on their basal metabolic activity" the authors investigate metabolites produced by pig sperm. They also try to address the sperm metabolic pathway used for energy production and whether the energetic metabolic state of sperm is related to their quality and function, including fertilisation and embryo development. The topic is relevant; the source of ATP (glycolysis vs. Oxphos) in sperm is under constant controversy and, although it is accepted that it may be specie-specific, much of the data remains debated and depend on different experimental approach.

In this manuscript, the authors investigate the issue through a metabolomic tactic, which is not so widely used in the field.

I should observe that results and interpretation are based on a statistic model and in the case of PCA there is some arbitrary use of sperm parameters (e.g; the indicators of sperm function and sperm quality are arbitrarily chosen). In my point of view, the parameters and the chosen methods to assess sperm function have less strength than those of sperm quality (motility and morphology, which are more universally standardized) so it may exist a bias in the results.

These issues are not necessarily something wrong but they may be explained.

I would appreciate it if the authors explain some of the following issues and clarify them in the text:
Title,

the authors should mention that the experiments are performed on boar sperm

In the Statistic section: The experiments are well designed but since I consider statistics is a main issue, could you please explain the statistical method used to get the accurate number of samples?

Results

The authors found some correlation between sperm quality and Krebs intermediates (lines 144 -151) that are not (to the point of view of this reviewer) deeply analyzed in the discussion section

Discussion

Lines 180-181: "sperm contain mitochondria with few and practically nonvisible inner membrane crests" is not completely exact. There is a recent publication where the ultrastructure of mitochondrial porcine sperm is shown by cryo-EM (doi.org/10.1073/pnas.2110996118). Moreover, the authors show that pig and horse sperm mitochondria have more expanded cristae and more condensed matrices than mouse sperm mitochondria.

Line 223. I found a contradiction to include the measurement of calcium in the function block and then postulate that it is as indicative of poor sperm quality.

Reviewer #2 (Remarks to the Author):

This is a review of "sperm physiology and in vitro fertility outcomes rely on their basal metabolic activity". The authors describe a study where swine semen samples were split and evaluated for in vitro sperm fertility parameters (motility, normality, acrosome), in vitro fertilization and embryo development, and for metabolic measures. The authors used analysis measures to link the fertility measures with the metabolic indicators. They reported certain indicators associated with well with measures for sperm fertility and others indicators poor fertility. The metabolic pathways of focus

concentrated on use of glycolysis or Oxphos, and that each of the metabolic indicators reflected on the sperm fertility status.

Overall an interesting and well written paper, although I am not as familiar with some of the approaches used in data presentation and analysis. To that end, I will suggest some changes that could help readers that are less versed in these analytical approaches and displays.

Major comments

- I think the authors may need to add some discussion on how sperm utilize the components in the extenders and media for energy. There is no mention of what is in the media or semen extender and what effect this may have on how sperm metabolize.
- Do the components on the uterine or oviductal fluids alter porcine sperm metabolism. What are energy components in vivo?
- How old were semen samples
- When was the study performed?
- Why would some sperm perform Oxphos if glycolysis pathway functional (i.e. why are some sperm compromised but others not in the same sample?)

Abstract

Does not really provide enough information to define what was tested and performed.

No mention of how many boars, samples, and methods.

Specific comments

L 24 instead of interrogate, use investigate (and elsewhere in paper).

L 31 confusing (greater IVF) as line 30 indicated fertilization not affected, but isn't fertilization part of IVF? Suggest rewording and not using IVF.

L 187 These mechanisms

L 229-30 is the format of the references correct for this journal (2011)

L 307 ..and function and the second was used for IVF experiments.

L 308 ..of embryo development at..

Figure 1. Can authors add a few sentences to instruct readers as to how to interpret graphs. The x and Y scales are all different for the graphs. It is not clear what was used to create the graphs (replicates, observations, etc.)

Figure 2. This is interesting, but not very helpful with so many colors, small words, and faint colors. I wonder if another type of diagram with a link to metabolic pathways would be more helpful and enlightening.

Table 1. Can the authors add subscripts or more information on how to interpret the column measures (i.e. FDR is this a false discovery rate as %?). What is a recursive rank and what does this tell us? What does a LR tell us?

Supplementary Table 1 .

What are the units for the mean? %, numbers? Not clear in this table.

Table 2.

The components 1-2 should be defined in a subscript for table

Table 3.

How do we interpret a similarity score? What does this mean

Reviewer #3 (Remarks to the Author):

The manuscript "Sperm physiology and in vitro fertility outcomes rely on their basal metabolic activity" aimed 1) the metabolic pathway preferentially used by porcine sperm for energy production; 2) whether the energetic metabolic state of sperm is related to their quality and function, and 3) the potential relationship between sperm energetic metabolism and oocyte fertilisation and subsequent embryo development.

In my opinion this is a classic example of a good paperwork. The whole manuscript is complete and intelligible.

It is stated in the Methodology section that "Sixteen ejaculates that met the quality standards (each came from a separate boar; i.e., 16 boars) were included in the present study and split into three aliquots." This seem a small number of samples, even, only one ejaculate per animal. Therefore, the selection of the samples (studies) to be analyzed may resulted in a bias in the Results and in spurious conclusions.

Responses to Reviewers' comments

Reviewer #1 (Remarks to the Author)

Reviewer (R): In the manuscript “Sperm physiology and in vitro fertility outcomes rely on their basal metabolic activity” the authors investigate metabolites produced by pig sperm. They also try to address the sperm metabolic pathway used for energy production and whether the energetic metabolic state of sperm is related to their quality and function, including fertilisation and embryo development. The topic is relevant; the source of ATP (glycolysis vs. Oxphos) in sperm is under constant controversy and, although it is accepted that it may be specie-specific, much of the data remains debated and depend on different experimental approach. In this manuscript, the authors investigate the issue through a metabolomic tactic, which is not so widely used in the field.

I should observe that results and interpretation are based on a statistic model and in the case of PCA there is some arbitrary use of sperm parameters (e.g; the indicators of sperm function and sperm quality are arbitrarily chosen). In my point of view, the parameters and the chosen methods to assess sperm function have less strength than those of sperm quality (motility and morphology, which are more universally standardized) so it may exist a bias in the results. These issues are not necessarily something wrong but they may be explained.

Authors (A): We would like to thank the reviewer for their helpful feedback. Certainly, sperm variables were chosen by authors. Related to this, we would like to emphasise that conventional sperm quality parameters (such as motility and morphology) were included in the PCA as the “sperm quality” block, together with the percentage of viable sperm. The rationale behind including other parameters in the “sperm function” block was based on the ongoing debate in the field. Indeed, several authors previously pointed out that traditional semen quality assessment does not provide a complete picture of sperm physiological status (i.e. 10.5713/ab.21.0072 or 10.1016/S0378-4320(01)00160-9). This clearly supports that other variables, such as the sperm function ones, need to be considered.

Moreover, it is worth clarifying that we ran three different PCAs in order to not mask the results of the sperm-function block with variables from the sperm-quality one; therefore, one should regard both results as complementary rather than scrambled. Considering that the purpose of the present work was to evaluate the relevance of cell metabolism on sperm physiological status (which compromises quality and function), we decided to take into account all the parameters we were able to analyse for each sample. Yet, this has been clarified in the Materials and Methods section, lines 380-389 and lines 515-538. Loading values of each variable from all PCA-vector-scores have been included in Supplementary Table 1.

R: I would appreciate it if the authors explain some of the following issues and clarify them in the text:

Title, the authors should mention that the experiments are performed on boar sperm.

A: We have added the species name in the title, following the reviewer's request.

R: In the Statistic section: The experiments are well designed but since I consider statistics is a main issue, could you please explain the statistical method used to get the accurate number of samples?

A: We thank the reviewer for this comment. The "pwr.f2.test" function of the "pwr" R package was used to determine the minimum sample size for linear regression. Considering an effect size of 35 % (parameter "f2" in the formula), which is based on previous metabolomics studies (i.e. 10.1016/j.jpba.2021.114450, 10.3390/ijms23063219), an alpha error of 0.05 ("sig.level" in formula) and a minimum power of the test of 60 %, we estimated a minimum degree of freedom of 14.13 (equivalent to $14.13 + 1 = 15.13$ observations). Therefore, we analysed 16 samples, which is suitable deliver robust results (as explained in lines 515-518).

In addition, we would like to indicate that the sample size was similar to that of previous studies (i.e. 10.3390/biom10060906, 10.1016/j.theriogenology.2022.12.009, 10.1016/j.jprot.2022.104791, 10.1111/rda.14270, 10.1002/mrd.23354, 10.1071/RD20304.). As this work evaluated, for each sample, a high number of

parameters (including motility using CASA system, several functionality parameters using flow cytometry and *in vitro* fertilization outcomes after *in vitro* oocyte maturation, *in vitro* fertilisation and *in vitro* embryo culture), and this entailed a significant amount of tests, we included the maximum number of animals that we could assume based on the lab work. This number of samples, which, as aforementioned, took into consideration the minimum sample size, was the one we could afford from the logistic perspective.

R: Results: The authors found some correlation between sperm quality and Krebs intermediates (lines 144 -151) that are not (to the point of view of this reviewer) deeply analyzed in the discussion section.

A: We thank the reviewer for this comment. In order to satisfy this concern, we have added few sentences to lines 244-267 to address the reviewer's concerns. In addition, we have also discussed further the results from the "sperm function" block in lines 268-299.

R: Discussion: Lines 180-181: "sperm contain mitochondria with few and practically nonvisible inner membrane crests" is not completely exact. There is a recent publication where the ultrastructure of mitochondrial porcine sperm is shown by cryo-EM (doi.org/10.1073/pnas.2110996118). Moreover, the authors show that pig and horse sperm mitochondria have more expanded cristae and more condensed matrices than mouse sperm mitochondria.

A: Line 180-181. We would like to thank the reviewer for providing us this new, interesting perspective on mitochondrial crests. If we did not understand it wrong while reading the cited paper, horse and pig sperm are similar in their mid-piece size. However, the authors of this article stated that "*Horse sperm mitochondria have an expanded intermembrane space and a condensed matrix (Fig. 1 I and J). Mouse sperm mitochondria have an expanded matrix, with a narrow intermembrane space and thin cristae (Fig. 1 K and L). Pig sperm mitochondrial morphology is intermediate (Fig. 1 G and H), and although the mitochondrial matrix was dense, we could identify individual complexes that resembled ATP synthase on cristae of FIB-milled mitochondria (SI Appendix, Fig. S2 A and B), which was confirmed by subtomogram averaging (SI Appendix, Fig. S2B).*" From these lines, we interpret that the pig spermatozoon is in

between their mouse and horse counterparts in terms of matrix condensation. In addition, the authors of this paper even acknowledge that mouse and pig have similar metabolism *“Indeed, horse sperm are dependent on oxidative phosphorylation (39), whereas pig (16) and mouse sperm (40, 41) are thought to rely largely on glycolytic mechanisms.”* In a similar way, in the discussion, they also state that *“Our data also show that mitochondrial dimensions and cristae architecture vary across species (Fig. 1), providing possible structural bases for interspecific differences in mitochondrial energetics. Horse sperm mitochondria appear to take on a more condensed appearance than their counterparts in pig and mouse, which may correlate with increased reliance on oxidative phosphorylation in horse sperm”*. We have made reference to all this valuable information in the revised Manuscript, as few lines mentioning the differences in mitochondrial crests between pig and horse sperm have been included to provide the reader with a comparison between species (lines 225-227).

R: Line 223. I found a contradiction to include the measurement of calcium in the function block and then postulate that it is as indicative of poor sperm quality.

A: Thank you for bringing this matter to light. Solely by itself, the parameter “intracellular calcium levels” has no positive or negative implication, because calcium is involved in many physiological processes. Yet, and as we explained in the paragraph, since it is an indicator of sperm capacitation - a process that should not be elicited in a non-capacitating medium -, we hypothesise that high levels of intracellular calcium could indicate early capacitation events. High levels of spontaneous capacitated sperm would imply that, by the time of fertilization, less sperm would be in appropriate physiological conditions for oocyte fertilization. For this reason, we referred to these samples as of “poorer quality”. In order to address the reviewer’s concern, this has been clarified in lines 276-279.

Reviewer #2 (Remarks to the Author)

R: This is a review of “sperm physiology and in vitro fertility outcomes rely on their basal metabolic activity”. The authors describe a study where swine semen samples were split and evaluated for in vitro sperm fertility parameters (motility, normality, acrosome), in vitro fertilization and embryo development, and for metabolic measures. The authors used analysis measures to link the fertility measures with the metabolic indicators. They reported certain indicators associated with well with measures for sperm fertility and others indicators poor fertility. The metabolic pathways of focus concentrated on use of glycolysis or Oxphos, and that each of the metabolic indicators reflected on the sperm fertility status.

Overall an interesting and well written paper, although I am not as familiar with some of the approaches used in data presentation and analysis. To that end, I will suggest some changes that could help readers that are less versed in these analytical approaches and displays.

A: We would like to thank the reviewer for the constructive comments that definitely improved the quality of the Manuscript. Their comments are addressed in detail below.

Major comments

R: I think the authors may need to add some discussion on how sperm utilize the components in the extenders and media for energy. There is no mention of what is in the media or semen extender and what effect this may have on how sperm metabolize.

A: We agree with the reviewer that substrates present in the medium influence cell metabolism. Unfortunately, the exact composition of commercial extenders used to prepare pig seminal doses is under industrial protection from the manufacturer, and we are unable to determine whether and how that composition could ultimately affect sperm metabolism. Certainly, this is a limitation of all studies of this nature, as commercial farms follow an established, authorized procedure to prepare the samples, and we cannot alter that way of functioning. Acknowledging that and in order to address the reviewer’s concern, this issue has been mentioned in lines 219-221. It is likely that, although all semen samples used in this work were diluted in the same medium, the differences

between this study and others could be explained by the usage of distinct extenders. In the light of all the aforementioned, one should note that, in the context of pig breeding through artificial insemination (AI), using commercial rather than custom-made extenders is convenient for the applicability of the results presented here.

R: Do the components on the uterine or oviductal fluids alter porcine sperm metabolism. What are energy components in vivo?

A: It is quite likely that changes in the composition of uterine and oviductal fluids have a direct impact on sperm metabolism. For this reason, a succinct reference in the Introduction of the original Manuscript has been made (lines 73-79). Following the reviewer's comment, however, we advise that this issue could be addressed further. For this reason, the Discussion has been revised and this point has been included (lines 307-316).

R: How old were semen samples

A: Samples were used within less of 24 h post-collection, despite the medium being able to maintain them properly up to seven or more days at 17°C. The age of animals from where samples were collected has also been included (line 377-378).

R: When was the study performed?

A: We have added this information in line 374.

R: Why would some sperm perform Oxphos if glycolysis pathway functional (i.e. why are some sperm compromised but others not in the same sample?)

A: It is known that sperm cells are highly heterogeneous in terms of physiology, which could contribute to explain why not all spermatozoa reach the oviduct nor are they capable to fertilize the oocyte. One explanation for this heterogeneity could be related to their metabolism. Although we cannot give a definite answer to this question, our hypothesis is that sperm with poorest quality and functionality traits predominantly have an oxidative

metabolism. This oxidative metabolism is likely to be linked to higher ROS production, which is known to cause damage to sperm membrane and DNA. This is explained in lines 333-344.

R: Abstract: Does not really provide enough information to define what was tested and performed. No mention of how many boars, samples, and methods.

A: Thank you very much for drawing our attention into this. Following the reviewer's comment, this has been amended in the revised version of the Manuscript.

Specific comments

R: L 24 instead of interrogate, use investigate (and elsewhere in paper).

A: The Manuscript has been entirely revised, and this has been amended as requested (lines 28 and 150)

R: L 31 confusing (greater IVF) as line 30 indicated fertilization not affected, but isn't fertilization part of IVF? Suggest rewording and not using IVF.

A: This has also been revised and clarified. Thank you.

R: L 187 These mechanisms

A: We have made this correction as requested.

R: L 229-30 is the format of the references correct for this journal (2011)

A: This has been corrected in the revised version of the manuscript.

R: L 307 ..and function and the second was used for IVF experiments.

A: This has been modified following the reviewer's recommendation.

R: L 308 ..of embryo development at.

A: The word “development” has been added, as per the reviewer’s request.

R: Figure 1. Can authors add a few sentences to instruct readers as to how to interpret graphs. The x and Y scales are all different for the graphs. It is not clear what was used to create the graphs (replicates, observations, etc.)

A: Following the reviewer’s request, the legend of Figure 1 has been revised, and the following information has been added to the legend: *"In x-axes, PLS regression plots show original centred reduced-dimension feature-vectors from each principal component analysis block. Values predicted by the models are displayed on y-axes. The slope is defined by the prediction capability values (Q2) tested through permutation tests. Vertical lines in permutation test plots indicate the obtained Q2 values for each block.*

R: Figure 2. This is interesting, but not very helpful with so many colors, small words, and faint colors. I wonder if another type of diagram with a link to metabolic pathways would be more helpful and enlightening.

A: This figure is the main outcome of a multi-block data analysis, and we think that a diagram with metabolic pathways would not provide an actual “integration” of the results. We are willing to acknowledge that the plot may be difficult to understand without a more detailed legend. We selected complementary modern colours to distinguish each block, so different aesthetics mean different parameters all in one comprehensive figure. This is now detailed in the figure legend. We also chose the font size to fit the journal’s artwork, and Manuscript and figure letters were set to have the same size. That being said, and based on the reviewer’s comment, the legend of Figure 2 has been revised (see below). Moreover, the unfolded pair-wise similarity scores of the multi-block data integration approach (which are shown in Figure 2) are also now provided in Supplementary Table 3. The following text has been added: *“Integrative network graph depicting correlations derived from N-integration with projection to latent structures between blocks. In vitro fertilisation outcomes (blue), metabolomics (red), sperm function (green) and sperm quality (orange) were found to be associated (similarity score > |0.3|; n = 16). Line*

colours are related to similarity scores: positive associations are in purple, whereas inverse associations are in yellow. Nodes (circles) represent variables and are sized according to the number of connections. Further information can be found in Supplementary Table 3. Abbreviations: col, colour; IVF, in vitro fertilisation; OH, hydroxy.”

R: Table 1. Can the authors add subscripts or more information on how to interpret the column measures (i.e. FDR is this a false discovery rate as %?). What is a recursive rank and what does this tell us? What does a LR tell us?

A: We are grateful to the reviewer for indicating that the Table needs further self-explanation. The following text has been added to the table caption:

“^aRecursive rank of double cross-validation PLS regression. In each block, metabolic markers were repeatedly ranked in each outer iteration and cumulated in the recursive rank feature. Only significant variables are numbered. ^bLoading rank displays absolute ranking of variables based on importance. ^cP-values and Betas from linear models. ^dFDR: false discovery rate. Corrected P-values utilizing the Benjamini–Hochberg formula.”

We also reworded the corresponding sentence in the statistical analysis section (LL541-542: *Only FDR-corrected P-values lower than 0.05 were considered statistically significant.*

R: Supplementary Table 1. What are the units for the mean? %, numbers? Not clear in this table.

A: We are grateful that the reviewer noticed this detail. This has been corrected in the revised version of the Manuscript. Indeed, thanks to the reviewer’s comment, we have realised that there was an outlier in one of the variables of the sperm function block. We have performed a new PLS analysis for this sperm function block, without this outlier, and both the performance of the model and the N-multiblock analysis have improved considerably the robustness (see the revised results and discussion sections).

R: Table 2. The components 1-2 should be defined in a subscript for table

A: We have reworded and added the following text to the caption of Supplementary table 2: *“Components 1 and 2 corresponded to the latent components of N-multiblock data analysis, utilizing in vitro fertilisation (IVF) outcomes as a reference dataset.”*

R: Table 3. How do we interpret a similarity score? What does this mean?

A: We have reworded the caption of Supplementary Table 3 and added the following text: *“The unfolded pair-wise similarity matrix was obtained from the network (Figure 2). Similarity scores can be interpreted as correlation values.”*

Reviewer #3 (Remarks to the Author)

Reviewer: The manuscript “Sperm physiology and in vitro fertility outcomes rely on their basal metabolic activity” aimed 1) the metabolic pathway preferentially used by porcine sperm for energy production; 2) whether the energetic metabolic state of sperm is related to their quality and function, and 3) the potential relationship between sperm energetic metabolism and oocyte fertilisation and subsequent embryo development.

In my opinion this is a classic example of a good paperwork. The whole manuscript is complete and intelligible.

It is stated in the Methodology section that “Sixteen ejaculates that met the quality standards (each came from a separate boar; i.e., 16 boars) were included in the present study and split into three aliquots.” This seem a small number of samples, even, only one ejaculate per animal. Therefore, the selection of the samples (studies) to be analyzed may resulted in a bias in the Results and in spurious conclusions.

Answer: We would like to thank the reviewer for such a good opinion of our Manuscript.

We did not increase this number, as we purchased samples from an AI centre that only housed animals with high-quality semen traits. We thank the reviewer for bringing this to light and, to avoid confusions, we have corrected this sentence in the revised version of the manuscript (LL377-379).

On the other hand, there are several reasons why we only included one ejaculate per animal. First, considering we sought to evaluate the sperm metabolism and semen quality and fertility traits of each boar, we advised it was important to do such an evaluation for the same ejaculate sample. The high number of techniques conducted made it difficult to include a high number of semen samples (thus, different boars) per each experimental week (we did 5-6 boars per experimental week). Second, it is known that semen quality can fluctuate seasonally. For this reason, we prioritized carrying out all the experiments in a short period (all experiments were carried out during June and July of 2021; information added in L374) to mitigate any seasonal effect. Thus, we estimated that it was better collecting one sample per boar rather than collecting more than one per animal. This, in our opinion, did not prevent us from getting reliable data.

Regarding the reviewer's concern about sample size, we would like to clarify that we first calculated the minimum sample size, utilizing the "pwr" R package and taking into consideration the statistical approach (prediction through regression). As previously mentioned in our response to Reviewer 1, considering a minimum size effect of 35%, which was based on previous metabolomics studies (i.e. 10.1016/j.jpba.2021.114450, 10.3390/ijms23063219), and an alpha error of 0.05, we estimated that the minimum sample size was 15.13 samples. In addition, we would like to indicate that the sample size was similar to that of previous studies (i.e. 10.3390/biom10060906, 10.1016/j.theriogenology.2022.12.009, 10.1016/j.jprot.2022.104791, 10.1111/rda.14270, 10.1002/mrd.23354, 10.1071/RD20304.). As this work evaluated, for each sample, a high number of parameters (including motility using CASA system, several functionality parameters using flow cytometry and *in vitro* fertilization outcomes after IVF and IVC), and this entailed a significant amount of tests, we included the maximum number of animals that we could assume based on the lab work. This number of samples, which, as aforementioned, took into consideration the minimum sample size, was the one we could afford from the logistic perspective.

REVIEWERS' COMMENTS:

Reviewer #1 (Remarks to the Author):

In the revised version of the manuscript: "Sperm physiology and in vitro fertility outcomes rely on their basal metabolic activity " the authors address my and other reviewer previous considerations. The revised version: "Sperm physiology and in vitro fertilising ability rely on basal metabolic activity: insights from the pig model " has been improved compared to the old version and I think it is ready to be published.

Reviewer #3 (Remarks to the Author):

The enormous effort made by the authors to improve the manuscript is noteworthy.

Responses to Reviewers' comments

Reviewer #1 (Remarks to the Author)

Reviewer (R): In the revised version of the manuscript: “Sperm physiology and in vitro fertility outcomes rely on their basal metabolic activity” the authors address my and other reviewer previous considerations. The revised version: “Sperm physiology and in vitro fertilising ability rely on basal metabolic activity: insights from the pig model” has been improved compared to the old version and I think it is ready to be published.

A: We would like to thank the reviewer for their comments that allowed to improve the manuscript.

Reviewer #3 (Remarks to the Author)

R: The enormous effort made by the authors to improve the manuscript is noteworthy.

A: We would like to thank the reviewer the positive feedback on our manuscript.